# Traditional Olive Tree Varieties in Alto Aragón (NE Spain): Molecular Characterization, Single-Varietal Oils, and Monumental Trees

Alfredo Serreta-Oliván [1], Rubén Sancho-Cohen [1,2], Ana Cristina Sánchez-Gimeno [3], Pablo Martín-Ramos [1,4,*], José Antonio Cuchí-Oterino [1] and José Casanova-Gascón [1,5]

1 Escuela Politécnica Superior, Universidad de Zaragoza, Carretera de Cuarte s/n, 22071 Huesca, Spain; serreta@unizar.es (A.S.-O.); rsancho@unizar.es (R.S.-C.); cuchi@unizar.es (J.A.C.-O.); jcasan@unizar.es (J.C.-G.)
2 Grupo de Bioquímica, Biofísica y Biología Computacional—Instituto de Biocomputación y Física de Sistemas Complejos, Universidad de Zaragoza, C. Mariano Esquillor Gómez, Edificio I+D, 50018 Zaragoza, Spain
3 Departamento de Producción Animal y Ciencia de los Alimentos, Facultad de Veterinaria, Instituto Agroalimentario de Aragón—IA2 (CITA-Universidad de Zaragoza), C/Miguel Servet 177, 50013 Zaragoza, Spain; anacris@unizar.es
4 ETSIIAA, Universidad de Valladolid, Avenida de Madrid 44, 34004 Palencia, Spain
5 Instituto Agroalimentario de Aragón—IA2 (CITA-Universidad de Zaragoza), Escuela Politécnica Superior, Carretera de Cuarte s/n, 22071 Huesca, Spain
* Correspondence: pmr@uva.es

**Abstract:** Recovering minority olive tree varieties helps preserve genetic diversity and contributes to sustainable agriculture practices. The International Olive Council has recognized the importance of conserving olive tree genetic resources and the European Union's Horizon Europe program has identified the preservation of crop diversity as a priority for sustainable food systems. In the work presented herein, old olive groves in the province of Huesca (NE Spain), managed according to the traditional model, were surveyed, sampled, and analyzed using molecular characterization techniques (based on EST-SNPs markers). Twenty-nine new varieties were identified and deposited in IFAPA's World Germplasm Bank of Olive Varieties. In the first step towards their valorization, eight single-varietal oils from Alto Aragon varieties were produced and characterized, and their organoleptic properties were evaluated, paving the way for the production of differentiated quality oils. Furthermore, ancient olive trees were selected and 3D scanned to promote their protection as singular or monumental trees and for oleo-tourism purposes. The reported findings highlight the rich olive-growing heritage of this northernmost frontier of olive tree cultivation in Spain.

**Keywords:** EST-SNP; genetic resources; Huesca; LIDAR; monumental trees; *Olea europaea*; olive oil

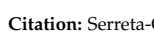



## 1. Introduction

Although it is currently considered a secondary crop, olive cultivation is deeply rooted in the territory of Alto Aragon (Huesca, Northeastern Spain), as it is part of the culinary culture and agronomic tradition in many localities. Olive cultivation in Aragon has been mentioned since ancient times, having developed from the Mediterranean coast and owing its expansion to the Greeks and Romans. The cultivated varieties changed over time, adapting to the rough and diverse territory, although classical literature mentions the oil more than the varieties. References to its quality have been numerous since the Middle Ages, as Aragon was an important oil producer since the 15th century [1]. During the Spanish Age of Enlightenment, several books mention the importance of the olive tree in this region. For instance, the economist and naturalist Ignacio Jordán de Asso traveled throughout Aragon and listed the localities that produced good oil and evaluated their production [2]. This author commented on the varietal exchanges between Seville and Aragon, the existence of plantations in the Pyrenees, and the goodness of the Empeltre

variety, without forgetting other Aragonese varieties, such as Royal, Negral, Racimillo, Acebuche, Picudillo, Manzanillo, Verdillo, and Zirujal. During the 19th century, the study of the varieties in Spain became important [3] and the varieties of Aragon began to be located. Megino-y-Metauten [4] mentioned that there were more varieties cultivated in the Kingdom of Aragon than in southern Spain, naming Royal, Negral, Sevillano, Racimillo or Uba, Azebuche, Picudo, Manzanillo, Verdillo, Cirujal or Largal, and Empeltre or Zuequecilla. In 1818, the military officer, naturalist, and engineer Félix de Azara described varieties from the Somontano (Huesca), such as Manzanilla (Alquecerana), Cerecera (Royeta), Negral (Neral), Panseña (Panseñera), Mochuta (Mochuto), Grosal, Olivonera, Verdeña, Minutesa, and Rebordenca [5]. Priego-Jaramillo [6] described many of the local varieties of Huesca, with a system of basic morphological identification and brief agronomic comments.

Nonetheless, since 1950, depopulation, intensive cereal mechanization, and the uprooting of olive trees [7] have reduced the cultivated hectares in Aragon to only 0.6% of the cultivated area at present [8]. Furthermore, olive cultivation has undergone important changes in recent years: on the one hand, its cultivation has shifted from the traditional producing regions to other areas where it was not grown (such as new irrigated areas in the southern part of the Alto Aragon); and, on the other hand, the number of varieties planted has been reduced. This reduction in the number of cultivated varieties is due to the fact that intensive olive-growing requires better-adapted varieties, which leads to the loss of genetic resources [9]. However, recovering minority olive tree varieties is important because it helps preserve genetic diversity and contributes to sustainable agriculture practices. It also recovers the landscape and marginal cultivation areas and optimizes the ecological footprint or biocapacity [10]. Consequently, the International Olive Council (IOC) has recognized the importance of conserving olive tree genetic resources and has established the Worldwide Olive Germplasm Bank of Cordoba, Spain (WOGBC) to collect and preserve diverse olive tree varieties. Similarly, the European Union's Horizon Europe program has identified the preservation of crop diversity as a priority for sustainable food systems. In addition, varietal recovery meets the goals of the FAO Committee on Agriculture on sustainable food systems and biodiversity integration in agricultural systems [11], and is oriented towards the 2030 goals [12], contributing to the achievement of 11 of the 17 SDGs (namely SDGs No. 1, 2, 3, 5, 8, 9, 10, 12, 13, 15, and 16).

Genotypic diversity within traditional olive tree varieties has been explored through various methodologies. Trujillo et al. [13] and El Bakkali et al. [14] conducted comprehensive analyses using microsatellite markers (SSR) and morphological traits to characterize germplasm banks in Marrakech (Morocco) and Córdoba (Spain). Gómez-Rodríguez et al. [15] employed SSR markers to genotype 36 cultivars within the core collection of the WOGBC. D'Agostino et al. [16] utilized Genotyping-By-Sequencing (GBS) to study a collection of 94 cultivars representing Italian olive germplasm. Studies on olive diversity have commonly utilized Single Nucleotide Polymorphism (SNP) panels [17], particularly Expressed Sequence Tag-SNPs (EST-SNPs), to explore the diversity within cultivated olive collections [18–20] and wild individuals [21]. In particular, the core set of 96 EST-SNP markers developed and validated by Belaj et al. [18] and Belaj et al. [19] has shown low genotyping error rates and a cost-effective genotype approach.

In the particular case of Aragon, most of the characterization and recovery studies of traditional varieties have been developed in the last 20 years [22]. Most agronomic studies have focused on the two most important varieties: Empeltre and Arbequina [23–26]. In contrast, the local or minority varieties recovered have hardly been studied [27,28]. However, it is recognized that these local varieties are a valuable source of diversity for breeding, given that they generally grow under difficult agroclimatic conditions [29].

In the province of Zaragoza, there is a varietal recovery project [27] as part of a Rural Development Program of the Sierra de Moncayo appellation. In the province of Teruel, several studies were carried out on the varieties grown in Bajo Aragon, including the morphological study of the stone of eight varieties by Puyuelo-Arilla [28]. Currently, the project "Oliveras Centenarias y Singulares, algo más sobre la historia de las Oliveras del

Bajo Aragón y Matarranya" is being developed, which has as a precedent a previous study [30]. On the other hand, the province of Huesca has received little attention. The most studied area is Somontano de Barbastro, where Verdeña is the main variety, followed by Empeltre, Blancal, Negral, and Arbequina. Viñuales-Andreu [31] recovered 18 varieties scattered throughout the region, which was the basis for subsequent studies on them. The recovered varieties have been planted in different parts of the territory, such as the Olearum Park (Barbastro) or the Buera Olive Forest (Huesca), as well as in experimental plots in several towns of the province. In the region of Hoya de Huesca, Viñuales et al. [32] located several varieties, which were described and characterized. In subsequent years, other regions were prospected, locating ten new varieties [33]. Thirty-two of these varieties were replanted in the Germplasm Bank of the Polytechnic School of the University of Zaragoza. The characteristics of the oils obtained from some of these varieties have been analyzed by Espada-Carbó et al. [34].

Building upon these initial efforts, this study had a dual focus: firstly, to identify and characterize minority olive tree varieties in the Alto Aragon, particularly in the Sobrarbe and Ribagorza regions; and secondly, to propose potential strategies for the valorization of this olive-growing heritage. To achieve the first objective, comprehensive surveys were conducted in old plantations and abandoned terraces, with the support of associations and landowners. Following this, genetic characterization utilizing 96 EST-SNP markers [18,19] was carried out on the identified specimens, in collaboration with the WOGBC. The second objective involved the preparation and characterization of single-varietal olive oils from selected varieties, alongside the identification and characterization of a subset of centenary olive trees with potential value as monumental trees.

## 2. Materials and Methods

### 2.1. Geographical Setting

The Alto Aragon, i.e., the northern part of the ancient Kingdom of Aragon, in the southern center of the Pyrenees, nowadays the province of Huesca, is a complex territory. The northernmost part is very rugged, reaching an altitude of over 3000 m, while the southern half is much flatter, with an average altitude of 400 m. Average annual rainfall ranges from over 1000 mm in the extreme north to 300 mm in the south. Consequently, in this territory, there is a wide mosaic of ecosystems.

### 2.2. Prospection for Local Olive Varieties in Old Plantations

Surveys were initially carried out in all the Alto Aragon. Farmers were asked for information on varieties that are common in their growing area, as well as minority and local varieties, filling out a form. The data collection forms were completed on-site in accordance with the guidelines set by international characterization organizations, specifically the IOC and International Union for the Protection of New Varieties of Plants (UPOV).

The results of the survey by localities are summarized in Figure 1, which shows the localities of origin of the plant material studied, and in Table S1, which shows the material found in each locality, with its traditional name, provided by the farmers. In some visits, sampling was not carried out, since the varieties found were modern commercial ones, or were well-known local varieties. In total, trees from 51 localities were marked, and 128 olive trees were selected for further study. Samples of some varieties were taken in several localities to check their diffusion throughout the province of Huesca.

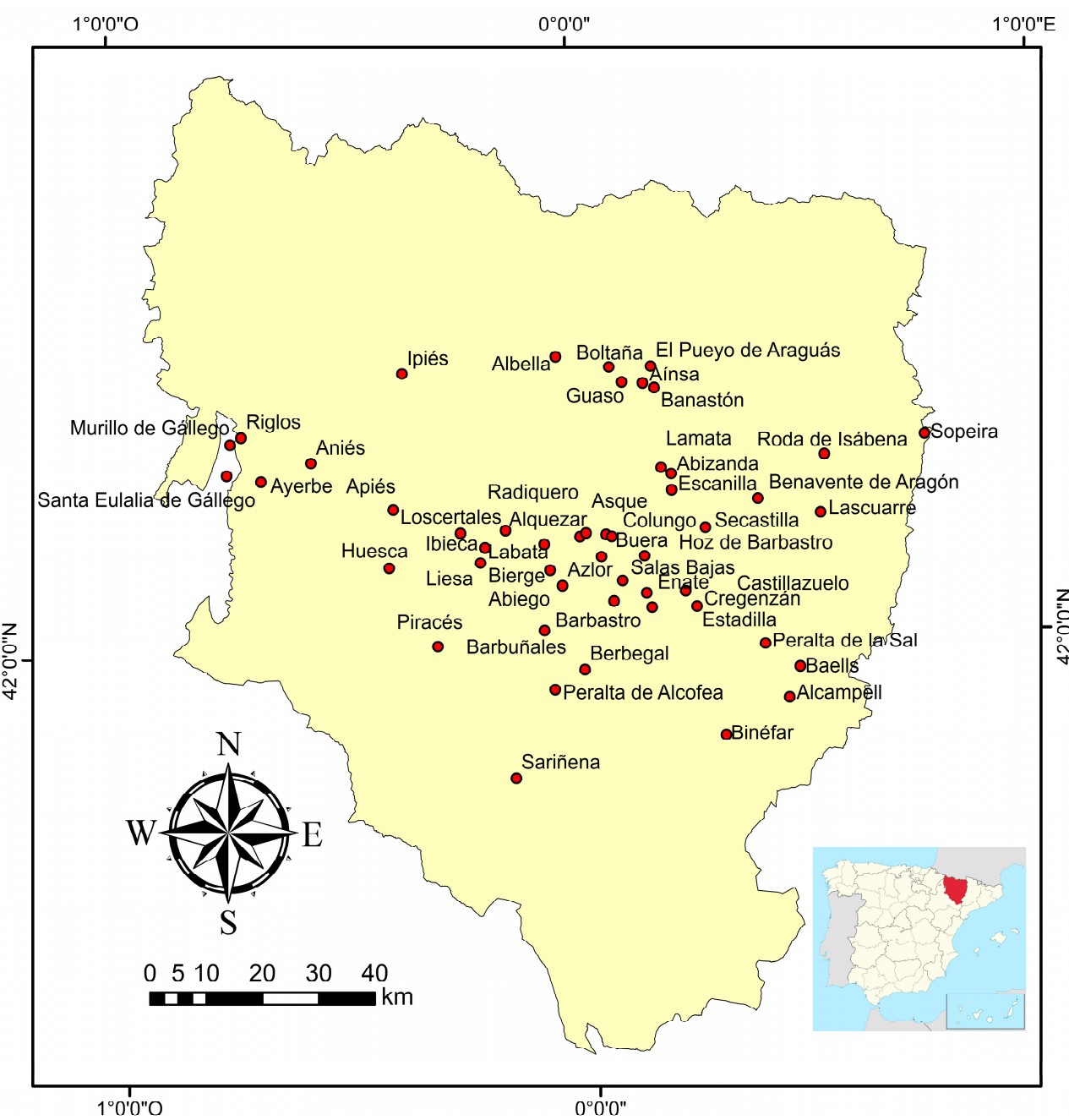

**Figure 1.** Map of the localities visited and from which olive trees were sampled or marked.

*2.3. Molecular Characterization*

A total of 92 samples of 56 varieties/cultivars from 39 locations throughout the Alto Aragon and 1 from Salamanca (Spain), added as a reference to know the discrimination efficiency of the method, were genotyped. Total genomic DNA was extracted from fresh leaves according to the cetyltrimethylammonium bromide (CTAB) method described by De-La-Rosa et al. [35]. DNA quantity and quality were estimated using spectrophotometry. A core set of 96 EST-SNP loci [18,19] was used to conduct the genotyping analysis. The amplification and SNP detection protocols are detailed in [19]. Two reference cultivars (Picual and Frantoio) were used as a positive control. Two genotyped samples, 'Royal de Calatayud_6' from Lamata and 'Royeta de Asque_4' from Radiquero, that showed at least four missing data (no call), were removed for downstream analyses. GenAlex 6.503 software [36,37] was used to conduct pairwise multi-locus matching analysis, using the codominant format, to detect redundant genotypes. A non-redundant data set, keeping



a unique representative genotype, was used to compute the genetic distance, which was analyzed to calculate the following key genetic parameters: number of non-redundant genotypes (N), average number of observed alleles (Na), number of effective alleles (Ne), Shannon's Information Index (I), observed heterozygosity (Ho), expected heterozygosity (He), and minor allele frequency (MAF) computed by GenAlEx v.6.503. Polymorphic information content (PIC) was computed by MolMarker v.1.0 software [38].

Principal coordinate analysis (PCoA), based on the genetic distance matrix of 58 non-redundant genotypes, was performed to elucidate the genetic relationships among olive varieties. The genetic distance and PCoA plot were computed and plotted using the "PCoA via Covariance matrix with data standardization" protocol available on GenAlEx.

The panel of 96 EST-SNPs of the 58 non-redundant genotypes was used to conduct the estimation of population genetic structure by admixture analysis using sparse nonnegative matrix factorization implemented in the LEA [39,40] R package. To delimit the number of clusters (K) to test, a preliminary admixture analysis was performed fixing only 10 repetitions and testing Ks from 1 to 58, the total number of non-redundant genotypes. According to the entropy criterion, Ks from 1 to 8 were deeply analyzed to compute the number of ancestral populations that best explained the genotypic data. Thus, the analysis was repeated testing the putative clusters (1 to 8) and conducting 100 repetitions. The cluster with the minimum entropy was determined as that which best explains the genotypic data. The Q-matrix of the best K and run was used to display a barplot for the ancestry proportions by pophelper R package and interactive shiny application pophelperShiny [41].

### 2.4. Single-Varietal Oils Preparation and Characterization

Aligned with the approach proposed by García-Vico et al. [42], the preservation of olive oil heritage is effectively pursued through the valorization of distinctive oils, facilitating the development of localized commercial circuits. Therefore, in this investigation, we crafted single-varietal oils from eight specific varieties originating in Alto Aragon. This selection comprised four previously recognized varieties (Alía, Blancal, Gordera de Abizanda, and Mochuto) and four newly identified ones (Minutera de Labata, Minutera de Viña, Olivonero de Ayerbe, and Rosal). The harvest, conducted in November 2022 at the University of Zaragoza's arboretum (Huesca, Spain), involved collecting 2–3 kg of each variety. Samples were taken to the Pilot Center for Food Science and Technology, where basic olive quality measurements were performed.

For the determination of the maturity index, 100 fruits were taken and the pigmentation of the epicarp and mesocarp of the olives was evaluated using a scale based on 7 levels of pigmentation and the formula provided in [9]. Concerning the oil extraction and calculation of fat yield, the oil was obtained using Abencor equipment (MC2, Seville, Spain) consisting of a mill, thermal blender, and centrifuge, simulating the industrial process. The olives were crushed using a hammer mill. The resulting paste was subjected to beating in a thermomixer at 26 °C for 30 min and then poured into the basket-type vertical centrifuge rotating at 3500 rpm and operated for 1 min. The centrifugation was repeated. The oily must was collected through the lower orifice into a graduated cylinder. After resting for at least 30 min, the oil volume was read and the yield was calculated.

Physical-chemical analyses of the oils were outsourced to the Agro-Environmental Laboratory of the Government of Aragon (Zaragoza, Spain), where acidity, peroxide index, ultraviolet extinction coefficients, total phenols, sterols, fatty acid profile, and oxidation stability were determined.

Concerning the sensory analysis, given that only small volumes of oil were available, it was carried out by an expert belonging to the Olive Oil Tasting Panel of Aragon. Standard blue-tasting glasses were used since color was not evaluated. About 14–16 mL of oil was placed in the glasses and covered with a watch glass. The sample was heated to 28 °C. Positive (fruity, bitter, and spicy) and negative (stale, rancid, vinegary, etc.) attributes were assessed on a scale of 0 to 10, according to [43].

### 2.5. Elaboration of a First Census of Singular Trees

As Lafuente-Benaches [44] suggests, the preservation of olive oil heritage extends to the valorization of centenary olive trees, recognizing them as monumental trees—integral components of biocultural heritage. These trees contribute significantly to the interpretation of historical landscape stratification and hold exceptional value in preserving the historical memory of the communities where they are situated.

The measurement method to define a monumental tree is described—for the Autonomous Community of Catalonia (Spain)—in Law 6/2020 [45], which establishes that the cataloging of monumental olive trees is from 350 cm of perimeter, measured at a height of 130 cm from the ground. Monumental trees, according to these criteria, were located in several localities. In this first census, individuals were selected from the localities of Guaso, Pueyo de Araguás, Riglos, and Santa Eulalia de Gállego. For their cataloging, the light detection and ranging (LIDAR) technique was chosen to obtain a complete scanned image of the olive trees, according to the procedure proposed by Hadas et al. [46]. The following measurements were recorded for each of the singular trees: location, perimeter at 130 cm above the ground, minimum perimeter height, base perimeter, maximum crown diameter, and crown volume (apart from other observations).

## 3. Results

### 3.1. Molecular Characterization

Genotypic analysis was conducted on 92 surveyed trees (refer to Tables S1 and S2), chosen according to specific criteria. A primary criterion was their classification as either an unknown variety or exhibiting uncertainties in visual characterization. Additionally, certain local varieties, previously identified years ago, were incorporated into the analysis because they were situated outside their recognized distribution areas. The analysis unveiled olive trees associated with 29 varieties not previously identified or classified in the WOGBC.

After the removal of two samples that showed at least four missing data (no call), the pairwise multi-locus matching analysis was applied to ninety samples, yielding fifty-eight non-redundant genotypes (see Table S3). Twelve redundant genotypes were identified among forty-four samples, with redundancy levels ranging from two to eight samples sharing the same genotype. All these redundant genotypes corresponded to identical varieties sampled from different locations.

Of the remaining 46 samples, each exhibited a unique and characteristic genotype. It is worth noting that samples 'Acebuche_1' and 'Acebuche_2', belonging to the same cultivar (Acebuche) and originating from the same location (Boltaña), and 'Buera_1' and 'Buera_2' (of Buera cultivar and location) displayed distinct genotypes (Table S3). It is also important to highlight that certain varieties, namely Minutera de Boltaña and Minutera de Labata, considered synonyms, exhibited different genotypes (refer to Table S3).

The analysis of genetic diversity encompassing 96 EST-SNP markers within the 58 non-redundant genotypes revealed a mean number of effective alleles per locus (Ne) at 1.78 (refer to Table S4). Minor allele frequency (MAF) values ranged from 0.07 to 0.49, with an average of 0.34. A substantial majority, 65 out of 96 EST-SNPs (67.7%), exhibited MAF values exceeding 0.30. Shannon's information index (I) displayed a range of 0.25 to 0.69, with a mean value of 0.62.

Examining heterozygosity, observed values (Ho) ranged from 0.14 to 0.76, averaging 0.48, while expected heterozygosity (He) fell between 0.13 and 0.5, with a mean value of 0.43. A noteworthy 77 out of 96 EST-SNPs (80.2%) demonstrated polymorphic information content (PIC) values exceeding 0.30 (see Table S4).

In the PCoA, the first two axes accounted for 10.32% and 8.04% of the total variance, respectively (refer to Figure 2). Owing to the nature of the study, it is not possible to ascertain the presence of a population structure among the studied genotypes associated with their geographical location. Nevertheless, certain trends in this regard are discussed in the subsequent section.

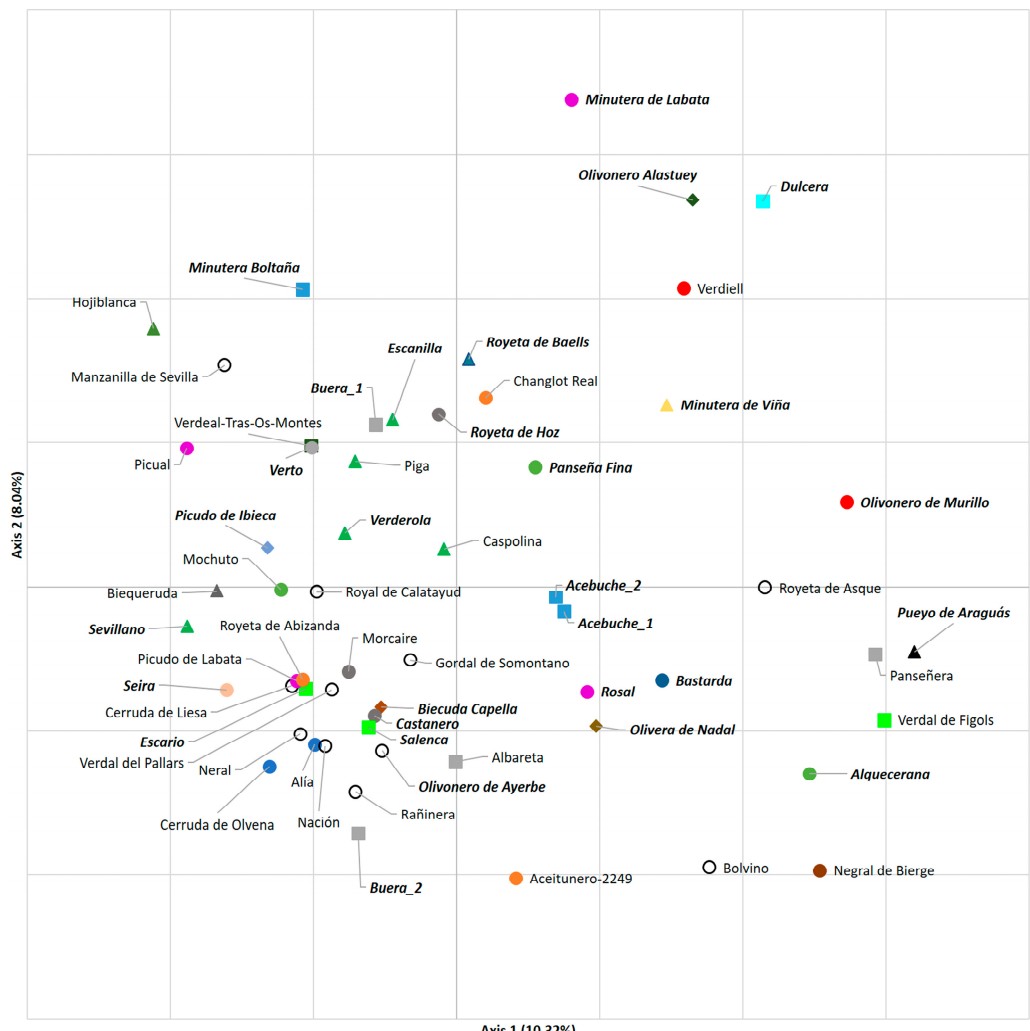

**Figure 2.** Principal coordinate analysis (PCoA) computed by GenAlEx v.6.503 via covariance matrix with data standardization using the genetic distance among 58 non-redundant genotypes based on 96 EST-SNP markers. Bold italic letters indicate new varieties. The same symbols and colors indicate the same locality of origin of the olive tree. The white circle was used as a generic symbol to indicate genotypes that include multiple samples from different locations. See Tables S1 and S2 for ID sample information.

The analysis of the population genetic structure identified K = 4 as the most probable number of clusters. By utilizing the admixture coefficients (Q-matrix) (refer to Figure 3a; Table S5), individual genotypes were assessed for their proportions of membership in each cluster (K1 to K4). The largest cluster, K1, encompassed 28 out of 58 genotypes, followed by K4 with 12, and K2 and K3 with 9 each.

Within each cluster, certain genotypes exhibited a limited ancestral mixture. For instance, genotypes such as Alía, Olivonero de Ayerbe, Cerruda de Liesa, Cerruda de Olvena, Gordal de Somontano, Nación, Neral, Rañinera, and Salenca in the K1 cluster; Hojiblanca and Manzanilla de Sevilla in K2; Dulcera, Minutera de Labata, and Olivonero Alastuey in K3; and Bolvino, Pueyo de Araguás, Negral de Bierge, Panseñera, and Verdal de Figols in K4. However, certain genotypes clustered in specific groups showed a remarkable ancestral mixture with two or more predominant groups. For example, Aceitunero-2249 and Acebuche_1, clustered in K1, displayed high ancestral proportions for both K1 and K4. In K2, Verto and Castanero exhibited similar ancestral proportions for K1, K2, and K4. Genotypes Royeta de Hoz, Buera_1, and Panseña Fina, clustered in K3, demonstrated high ancestral proportions for groups K1 and K3, and Royeta de Baells showed an affinity

for K2, K3, and, to a lesser extent, K4. In the K4 cluster, Caspolina and Olivera de Nadal showed ancestral proportions of K2 and K1, respectively (Table S5). This ancestral mixing of some genotypes was also detected in the PCoA, where genotypes with a high mixture of ancestral proportions were positioned in close proximity to genotypes from other groups (e.g., Castanero, Buera_1, Panseña Fina, and Royeta de Baells) (Figure 3b).

**(a)**

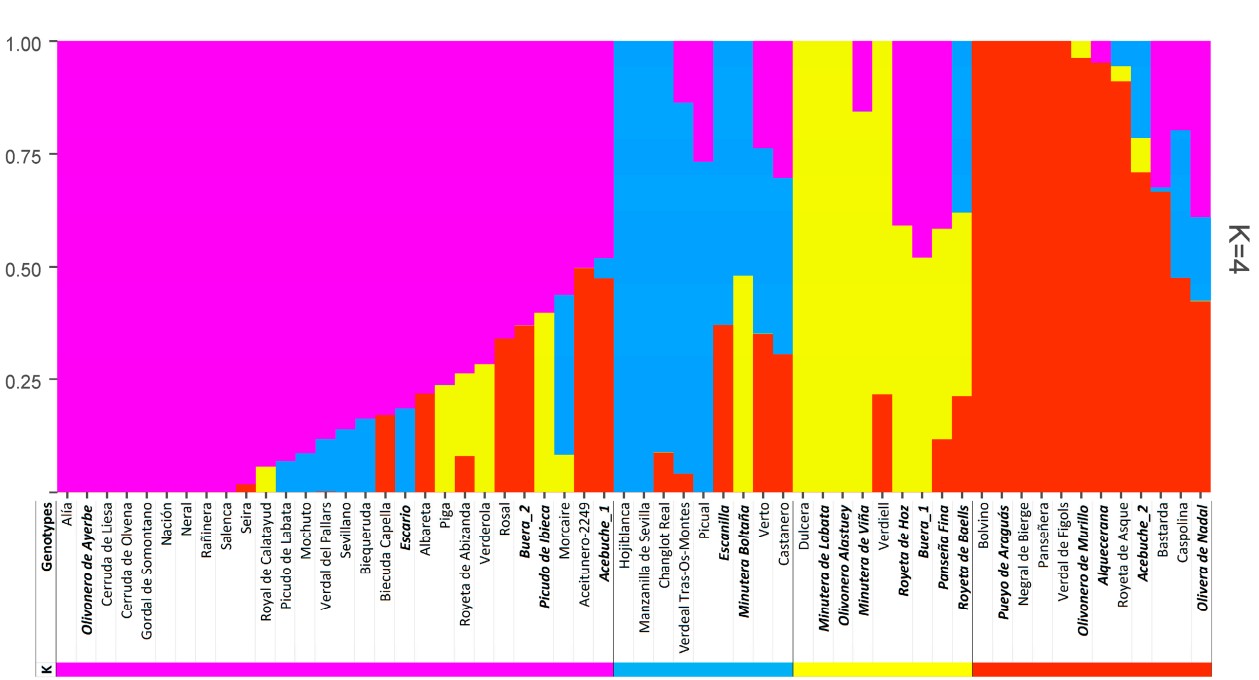

**(b)**

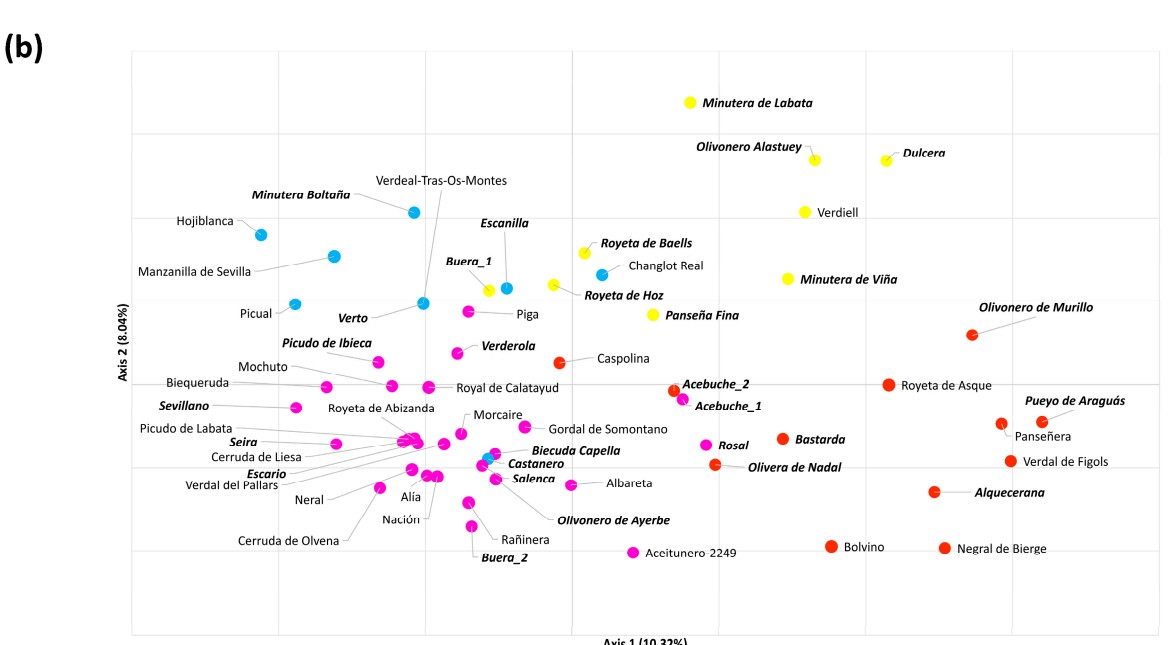

**Figure 3.** (**a**) Population structure representation by barplot based on the proportions of ancestry coefficients (Q-matrix; Table S5) using the 96 EST-SNPs panel of 58 non-redundant genotypes computed by LEA R package; (**b**) PCoA (refer to Figure 2) showing the genotypes colored according to the predominant ancestral group. K indicates the number of clusters. Bold italic letters indicate new varieties.

### 3.2. Characterization of Single-Varietal Oils

Table 1 shows the maturity index results for the eight selected olive varieties from Huesca with different genotypes. The choice of these varieties was guided by multiple criteria. Foremost was the oil quality, as indicated by the farmers who participated in the survey. Additionally, considerations included the biennial bearing patterns observed in specific varieties during the study year, the ample availability of olives for harvesting from the trees, the health condition of olives in certain varieties, and the project's budgetary constraint, which necessitated limiting the analyses to eight varieties. The Blancal variety showed the highest maturity index (5), followed by Minutera de Labata (4.9) and Gordera de Abizanda (4.3). The Rosal variety showed the lowest maturity index with 2.5 at the beginning of the veraison.

**Table 1.** Characterization of single-varietal oils obtained from eight varieties from the province of Huesca: olive maturity index, fat yield, physicochemical parameters, and sensory attributes.

| Variety | Maturity Index | Yield (%) | Physicochemical Parameters | | | | | | | | Sensory Attributes | | |
|---|---|---|---|---|---|---|---|---|---|---|---|---|---|
| | | | Acidity (%) | Peroxide Index (meqO2/kg) | K232 | K270 | Total Phenols (mg/kg) | Oleic Acid (%) | Sterols (mg/kg) | Oxidation Stability (h) | Fruity | Bitter | Spicy |
| Mochuto | 3.8 | 14.5 | $0.20 \pm 0.02$ | $3.5 \pm 0.5$ | $1.96 \pm 0.10$ | $0.17 \pm 0.01$ | 1026 | $75.94 \pm 0.30$ | 1235 | 21.5 | 4.7 | 4.5 | 4.2 |
| Olivonero Ayerbe | 3.6 | 21.9 | $0.15 \pm 0.02$ | $4.7 \pm 0.7$ | $1.67 \pm 0.08$ | $0.10 \pm 0.01$ | 580 | $75.05 \pm 0.30$ | 1147 | 17.0 | 2.9 | 0.5 | 3.3 |
| Alía | 4.0 | 17.6 | $0.18 \pm 0.02$ | $3.5 \pm 0.5$ | $1.54 \pm 0.08$ | $0.10 \pm 0.01$ | 451 | $76.47 \pm 0.31$ | 1484 | 17.3 | 5.8 | 1.4 | 3.3 |
| Blancal | 5.0 | 22.8 | $0.16 \pm 0.02$ | $2.1 \pm 0.3$ | $1.83 \pm 0.09$ | $0.14 \pm 0.01$ | 800 | $66.19 \pm 0.26$ | 1645 | 13.7 | 3.3 | 4.0 | 3.7 |
| Rosal | 2.5 | 16.5 | $0.16 \pm 0.02$ | $4.5 \pm 0.7$ | $1.73 \pm 0.08$ | $0.13 \pm 0.01$ | 653 | $79.22 \pm 0.32$ | 1266 | 17.6 | 4.3 | 0.4 | 3.7 |
| Gordera Abizanda | 4.3 | 8.1 | $0.17 \pm 0.02$ | $1.7 \pm 0.3$ | $1.62 \pm 0.08$ | $0.13 \pm 0.01$ | 753 | $75.96 \pm 0.30$ | 836 | 24.4 | 3.7 | 1.9 | 2.5 |
| Minutera Viña | 3.6 | 17.1 | $0.22 \pm 0.02$ | $4.2 \pm 0.6$ | $1.74 \pm 0.09$ | $0.16 \pm 0.01$ | 864 | $78.10 \pm 0.31$ | 902 | 29.9 | 2.2 | 1.6 | 3.2 |
| Minutera Labata | 4.9 | 10.3 | $0.13 \pm 0.01$ | $2.2 \pm 0.3$ | $1.64 \pm 0.08$ | $0.12 \pm 0.01$ | 691 | $76.99 \pm 0.31$ | 974 | 30.7 | 2.8 | 1.0 | 3.9 |

The different varieties showed notably different yields in oil extraction. The variety that showed the highest yield was Blancal, with 22.8%, probably related to a higher maturity index. The Olivonero de Ayerbe variety also had a high oil extraction yield (21.9%), even with a lower maturity index. In turn, the Gordera de Abizanda variety had the lowest yield in oil extraction (8.1%), which can be ascribed to the high stone content in these fruits.

All the physicochemical parameters (acidity, peroxide index, $K_{232}$, and $K_{270}$) for the oils were within the limits established by European regulations for the extra virgin olive oil category [47]. Mochuto and Minutera de Viña had a very high total phenol content, which is relevant given that they are natural antioxidants. The oleic acid content ranged from 66.19% for the Blancal olive oil to 79.22% for the Rosal olive oil. Concerning total sterols, the Blancal olive oil stood out because of its very high content. As for the oxidation stability, it was approximately 30 h in Minutera de Labata and Minutera de Alquézar, a result consistent with their high total phenol and oleic acid contents.

Concerning the sensory analysis, important differences were detected among the varieties. The olive oil with the best sensory quality was that of the Alía variety, with a higher fruitiness with floral notes and low values of bitterness and medium spiciness. The oil of the Mochuto variety also had a high fruitiness with banana notes, albeit with higher values of spiciness and bitterness. The oils of the Blancal and Olivonero de Ayerbe varieties showed small staling defects. In the first case, this could be due to the high degree of ripening of the olives and, in the second case, because the oil was unfiltered. Nonetheless, more sensorial analysis would be necessary to confirm these findings. Other oils would be classified as extra virgin olive oils.

### 3.3. Census of Singular Trees

Regarding the second approach for valorizing the olive tree heritage, Figure 4 illustrates outcomes derived from the application of LIDAR-based millimeter-precision scanning for the cataloging of the singular trees. In contrast to the time-consuming conventional manual characterization process, this technique facilitated the creation of a digital twin at the time of data collection in approximately 2 min. This digital twin can be revisited or utilized for the straightforward extraction of morphological characterization parameters (refer to Figure S1 and Table 2).

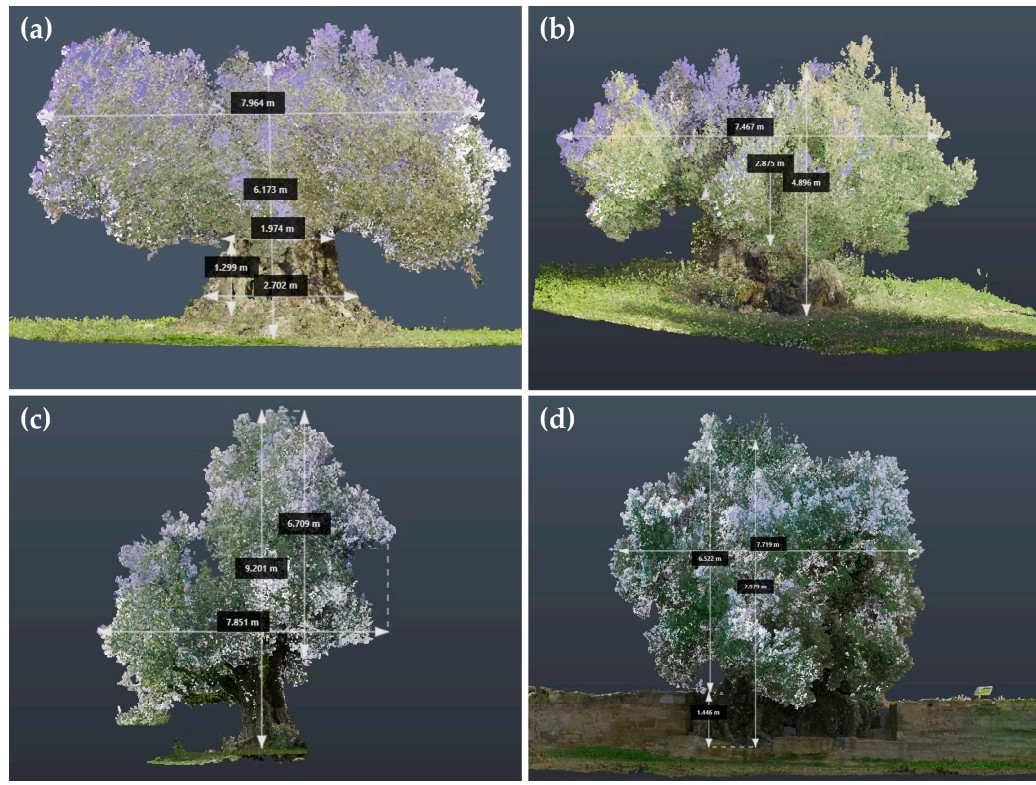

**Figure 4.** Examples of LIDAR scanning of monumental trees at the locations of (**a**) Santa Eulalia de Gállego, (**b**) Riglos, (**c**) Guaso, and (**d**) Colungo.

**Table 2.** Perimeters and areas of the sections at 1.30 m above the soil level, together with age estimations, for the examples of monumental trees scanned with LIDAR presented in Figures 4 and S1.

| Location | Variety | Perimeter (m) | Area (m$^2$) | Diameter (mm) | Age | | | |
|---|---|---|---|---|---|---|---|---|
| | | | | | Michelakis [47] | Pannelli et al. [48] | Arnan et al. [49] | Koniditsiotis [50] |
| Santa Eulalia de Gallego | New variety | 10.78 | 2.26 | 2540 | 1693 | 1600 | 790 | 300 |
| Riglos | Rañinera | 10.07 | 3.68 | 2860 | 1907 | 1801 | 890 | 370 |
| Guaso | Rañinera | 4.25 | 0.59 | 1350 | 900 | 850 | 420 | 213 |
| Colungo | New variety | 16.15 | 5.52 | 3350 | 2233 | 2110 | 1042 | 533 |

The results of the analysis on the age of centennial olive trees using various growth rate formulas are also presented in Table 2. Various equations, based on radial growth rates, the radius at a height of 1.0 m in cm, the diameter at a height of 1.3 m in cm, and the

perimeter in meters, were used. These equations, derived from studies by Michelakis [47], Pannelli et al. [48], Arnan et al. [49], and Koniditsiotis [50], allowed us to estimate the age of the ancient olive trees.

## 4. Discussion

### 4.1. Genotypic Diversity of Traditional Olive Tree Varieties

The core set of 96 EST-SNP markers, as validated by Belaj et al. [18] and Belaj et al. [19], has proven efficient in detecting redundant and new genotypes in the present study. In our multi-locus analysis, we discovered that 44 out of the 90 samples studied exhibited at least one sample with the same genotype. Redundant genotypes, even among different accessions of olive germplasm banks, have been consistently identified in various studies [13,19,51]. Among the non-redundant genotypes, 29 belong to varieties not included in the WOGBC. This set of previously undiscovered traditional olive varieties represents valuable genetic resources for olive breeding programs, particularly noteworthy for their adaptation to challenging agroclimatic conditions.

The diversity values (Ne, MAF, Ho, He, I, and PIC) recovered in the present study, calculated using a non-redundant genotype size between 56 and 58, closely mirrored the average values reported by Belaj et al. [19], who used a non-redundant genotype size between 644 and 668.

PCoA showed percentages of variation closely resembling those reported by Belaj et al. [19], who recorded values of 10.99% (axis 1) and 7.73% (axis 2). However, the limitation in the number of individuals, varieties, and geographical distribution in the sampling introduces a bias concerning the imputed population structure. Previous studies have reported population genetic structures that are, in some cases, partially associated with the geographical distribution of olive varieties/cultivars, conducted across various spatial scales [16,19,20,51,52]. In the investigation by Miazzi et al. [53], a comprehensive study of 177 minor genotypes (rare and ancient olive germplasm) sampled in the Apulian region (Italy) was carried out. The study encompassed morphological, molecular (microsatellite markers), technological, and phytosanitary status analyses, comparing the minor genotypes with reference cultivars. The population structure analysis revealed a distinct separation between the national Italian germplasm and the locally examined varieties. Conversely, studies like the one conducted by Biton et al. [17] grouped cultivars based on their purpose (oil, table, or double purpose) rather than their geographic origin.

The correlation between population structure and physicochemical parameters or sensory attributes of the oil was not explored in the present study. Instead, it emphasizes the characterization of traditional varieties and singular olive trees situated in the Alto Aragon region, employing diverse approaches such as morphological, oil, and genotypic analyses. The study does not delve into phylogenetic aspects either, as this would necessitate a more extensive sampling of varieties and the inclusion of reference varieties. Nevertheless, it is plausible to hypothesize about certain genotypic relationships and putative population structures within the sampled area. The genotypes clustered in the largest group, K1, exhibit a predominant West–East geographical distribution. Group K2 encompasses genotypes located from North-East to South-East. Groups K3 and K4 cluster genotypes from the Center and East, but K3 also includes genotypes from the South-East, contrasting with K4, which comprises genotypes from the North-East. Several varieties, such as Hojiblanca, Picual, and Manzanilla de Sevilla—traditionally widespread in olive groves in southern Spain (Andalusia and Extremadura), along with Changlot Real in eastern Spain (Valencia) [3], and Verdeal Tras-Os-Montes in Portugal [54]—are grouped together with minority varieties from Alto Aragon, namely Escanilla, Minutera Boltaña, Verto, and Castanero, in cluster K2 (Figure 3a,b). However, it is noteworthy that these Alto Aragon varieties exhibit a pronounced ancestral mixture in their proportions (Table S5), unlike their counterparts cultivated in the southern and eastern regions of Spain mentioned above.

Overall, the genotypic characterization conducted in this study has proven valuable in identifying redundant and novel genomes among traditional olive tree varieties in Alto



Aragon. This contributes significantly to the expansion of plant genetic resources available for the implementation of olive breeding programs.

### 4.2. On the Single-Varietal Oils

The eight single-varietal olive oils, except for Blancal and Olivonero de Ayerbe due to minor defects identified in the sensory analysis, met the classification criteria for extra virgin olive oil according to the physicochemical parameters and sensory attributes specified in European regulations [55]. However, notable differences were observed in both the composition of the fruits and the resulting olive oils. The maturity index, ranging from 2.5 (indicating skin color with the fruit surface turning red, purple, or black) for Rosal to 5 (denoting skin color all purple or black with less than half the flesh turning purple) for Blancal olives, significantly influenced olive oil yield, which varied from 10.3% to 22.8%, and quality [56]. Higher maturity indices were associated with increased olive oil yield. In the case of Minutera de Labata, the lower yield could be attributed to the large size of the stone.

Substantial variations were also noted in the total phenol content of the single-variety olive oils, ranging from 451 mg/kg for the Alía variety to 1026 mg/kg for Mochuto (more than twice the amount). These values surpass those of other varietal olive oils in Aragon, such as Royal de Calatayud, Negral de Sabiñán, Arbequina [57], or Empeltre [58]. Oleic acid, a crucial component of olive oil, exhibited a range from 66.19% to 79.22%. The lowest content was found in the olive oil from the more ripe olives (Blancal), while the highest was from the less ripe olives (Rosal). This latter content was similar to that described in previous works for olive oils from Aragon of Racimilla [27] and Verdeña [59] varieties. Additionally, differences in total sterol content were observed among the eight single-variety olive oils, with most exceeding 1000 mg/kg, the limit established by European regulations for the purity characteristics of extra virgin olive oil [55]. Blancal olive oils exhibited the highest sterol content, while Gordera de Abizanda, Minutera de Alquézar, and Minutera de Labata remained within the specified limits.

Remarkably, oxidation stability varied significantly, with Minutera de Labata demonstrating very high stability (30.7 h) and Blancal olive oils exhibiting very low stability (13.7 h). In all cases except Blancal, the values exceeded those reported for other single-variety olive oils from Aragon [57], with Empeltre showing lower values (10.03 h) [58].

All the olive oils exhibited positive sensory attributes, including fruity, spicy, and bitter notes, except for Blancal and Olivonero de Ayerbe, where small defects were detected. Notably, Alía olive oil stood out as the most fruity, while the Mochuto variety oil displayed the most pronounced bitter and spicy characteristics, correlating with its high total phenol content. No previous studies on the sensory properties of these single-variety olive oils have been documented.

Considering both chemical composition and sensory quality, the production of unique single-varietal olive oils could be a valuable strategy for the valorization of these indigenous olives. From a nutritional perspective, Mochuto monovarietal olive oils could be particularly interesting due to their high phenol content, while Alía could be an appealing option for its distinctive sensory properties. Additionally, blending these olive oils could offer the opportunity to combine the best nutritional and sensory attributes.

### 4.3. On the Protection of Monumental Olive Trees

The valorization of monumental olive trees is taking place throughout the Mediterranean basin, as highlighted by previous studies conducted in Albania [60], Cyprus [61], Lebanon [62,63], Portugal [64], Italy [65–71], Israel [72], Montenegro [73], Crete [74], Malta [75,76], and Spain [77].

In the case of Spain, the protection of monumental olive trees is recent and is not legislated by the Spanish Government, although it is regulated in a few Autonomous Communities, including Andalusia [78], Valencia [79], Catalonia [45], and Aragon [80]. In Andalusia, Gómez-Gálvez et al. [81] consider 'Singular Olive Trees' those specimens

that stand out in size (trunk perimeter and height), age, production, and aesthetic and landscape interest. This last aspect is an asset for sustainable development and regional competitiveness, which has even led to considering their heritage recognition at the highest international level [82].

In the particular case of Aragon, only some centenary olive trees are protected through the figure of "Árboles y arboledas singulares de Aragón" (tr. "singular trees and groves in Aragon") [80], which includes one of the olive trees under study, namely the "Olivera de Nadal" [83].

The finding of many monumental trees throughout the province of Huesca suggested their indexing and study, both for conservation purposes (under the 'singular tree' protection legislation), and as a means to promote olive-oil-related tourism in these rural areas (routes can be envisaged in the Somontano de Guara and the highest part of Sobrarbe and Ribagorza counties). This would imply replicating in the province of Huesca projects that have been successful in other regions: e.g., the project "Apadrinounolivo.org", which aims to recover abandoned olive trees with funding from private donors, financing the care of these olive trees and their recovery [84]; the project "Prospection of the monumental olive trees of Andalusia" [85], which locates and characterizes the aforementioned olive trees; and the "Sistema Agrícola Olivos Milenarios del Territorio Sénia" [86,87], recognized as an Important System of World Agricultural Heritage (SIPAM) by FAO, which not only protects olive trees but also recovers them, obtains oil and promotes olive oil tourism.

### 4.4. Limitations of the Study

Regarding the genotypic analysis, the limitation in the number of samples and the proximity between them, as the study is focused on the Alto Aragon region, does not allow conclusions to be drawn about its population structure. The small geographical scale, as well as the lack of information on morphological and physiological characters, limits the development of conclusions regarding the main driver in the population structure of the samples studied beyond the variety to which they belong. On the other hand, it would be necessary to expand the number of genotyped samples in which physicochemical and organoleptic parameters of their oils are analyzed to determine the influence that these attributes have or do not have on the population structure.

Concerning the single-varietal oils, it is crucial to note that future research is needed to confirm these preliminary results. Given that the eight varieties in this study were harvested simultaneously, variations in MI are likely to have affected extraction yield, physicochemical parameters, and sensory attributes. Subsequent studies should explore the comparison of varieties at a specific maturity index. Moreover, studying more crop years and increasing the volume of olive oil production for broader testing would contribute to the robustness of the findings. The limitations for the valorization of these single-variety olive oils stem from the relatively small cultivated area. To promote commercial single-varietal olive oils, the replantation of these varieties in different orchards is recommended.

As for the monumental olive tree census, it is important to acknowledge the existence of contradictory findings in the field of ancient olive tree age estimation. Recent studies by Camarero et al. [88], Cherubini et al. [89], Cherubini et al. [90], and Ehrlich et al. [91] have challenged some of the assumptions and methods used in previous research. Camarero et al. [88] have suggested that factors such as climate variations and soil conditions can significantly influence olive tree growth rates, potentially leading to overestimations of tree age when using certain formulas. Cherubini et al. [89] and Cherubini et al. [90] have emphasized the importance of dendrochronological techniques for more accurate age determination in ancient olive trees, highlighting the need for a multidisciplinary approach to age estimation, combining multiple methods to cross-verify results. Furthermore, Ehrlich et al. [91] have raised questions about the reliability of using linear growth rate equations without considering potential non-linear growth patterns in ancient olive trees.

In light of these contradictory findings, it becomes evident that estimating the age of centennial olive trees is a complex and evolving field of research. While the growth rate

equations we utilized provide a useful framework, they should be considered alongside the insights and cautions presented by Camarero et al. [88], Cherubini et al. [89], Cherubini et al. [90], and Ehrlich et al. [91]. Future research should aim to integrate these varied perspectives and methodologies to improve our understanding of these remarkable ancient trees' true ages and the factors influencing their growth.

## 5. Conclusions

The imperative to reduce inputs and adapt to climate change poses new challenges for future crops, underscoring the increasing necessity to incorporate novel sources of genetic variability for crop improvement. The genotypic analysis in this study identified 29 previously undiscovered traditional olive varieties in Alto Aragon, offering valuable genetic resources for olive breeding programs. The conservation of these traditional varieties emerges as crucial, serving as a reservoir of plant genetic resources. In an initial step towards their valorization, single-varietal olive oils were produced from eight of these varieties, with six meeting the criteria for extra virgin olive oil according to European regulations. Notably, Mochuto monovarietal olive oil stood out for its nutritional richness, boasting a high phenol content, while Alía olive oil presented distinctive sensory properties. As an alternative/complementary approach to valorization, numerous monumental trees were identified across the province of Huesca, suitable for conservation under the 'singular tree' protection legislation and as a catalyst for promoting olive oil-related tourism in rural areas, especially in the Somontano de Guara, Sobrarbe, and Ribagorza counties. These findings underscore the rich olive-growing heritage of Alto Aragon and its untapped potential for valorization.

**Supplementary Materials:** The following supporting information can be downloaded at: https://www.mdpi.com/article/10.3390/agriculture13122204/s1, Table S1: Results of the survey of plant material according to the different localities; the olive trees selected for DNA analysis; results of the analysis (BGMO); interesting varieties (new genotype) and observations; Table S2: Biallelic SNP set of 96 EST-SNP Markers for 92 samples and two positive controls; Table S3: Sorted Multi-locus Genotypes; Table S4: Diversity parameters of the 96 EST-SNPs genotyped in the 58 non-redundant genotypes; Table S5: Matrix of admixture coefficients (Q-matrix) for K = 4 indicating the ancestral populations of the 58 non-redundant genotypes; Figure S1: Examples of results from LIDAR scanning of the monumental olive trees: tree cross-sections at 1.3 m above the soil level.

**Author Contributions:** Conceptualization, J.C.-G.; methodology, A.S.-O., R.S.-C., A.C.S.-G. and J.C.-G.; software, A.S.-O. and R.S.-C.; validation, P.M.-R. and J.A.C.-O.; formal analysis, A.S.-O., R.S.-C., A.C.S.-G. and J.C.-G.; investigation, A.S.-O., R.S.-C., A.C.S.-G., P.M.-R., J.A.C.-O. and J.C.-G.; resources, J.C.-G.; data curation, R.S.-C.; writing—original draft preparation, A.S.-O., R.S.-C., A.C.S.-G., P.M.-R., J.A.C.-O. and J.C.-G.; writing—review and editing, A.S.-O., R.S.-C., A.C.S.-G., P.M.-R., J.A.C.-O. and J.C.-G.; visualization, A.S.-O. and R.S.-C.; project administration, J.C.-G.; funding acquisition, J.C.-G. All authors have read and agreed to the published version of the manuscript.

**Funding:** This research was funded by Diputación Provincial de Huesca through the Félix de Azara program.

**Institutional Review Board Statement:** Not applicable.

**Data Availability Statement:** The data presented in this study are available in the supplementary material.

**Acknowledgments:** The authors gratefully acknowledge technical and human support provided by SGIker (UPV/EHU/ERDF, EU); Raúl De la Rosa-Navarro and Angjelina Belaj (IFAPA, Centro Alameda del Obispo, Córdoba, Spain); and the World Olive Germplasm Collection of Córdoba (ESP-046).

**Conflicts of Interest:** The authors declare no conflict of interest. The funders had no role in the design of the study; in the collection, analyses, or interpretation of data; in the writing of the manuscript; or in the decision to publish the results.

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
