# Peer review of "Traditional Olive Tree Varieties in Alto Aragón (NE Spain): Molecular Characterization, Single-Varietal Oils, and Monumental Trees"

_agriculture, doi:10.3390/agriculture13122204_

Round 1

Reviewer 1 Report

Comments and Suggestions for Authors

This paper can be accepted for publications with minor modifications. The manuscript presents valuable results on olive research using suitable methods. The research also is original. Nevertheless, several modifications or additional information must be included in the final version of the manuscript (attached the PDF with revisions):

1.      Title can be improved, it reflects in this forms only the use of molecular markers.

2.      Several sentences need to be rephrased.

3.      The paragraph citing the objectives need to be rephrased, it is not clear in this present form.

4.      When you are citing the number of accessions, gentotypes, varieties sometimes it is not clear to follow example for samples one tine you cite 92 another 90….?????

5.      The first paragraph of results should be in material and methods

6.      Paragraph 3.4. should be in material and methods

7.      You have to clarify the choice of the 08 varieties used for the characterization of single varieties oils

8.      For the paragraph of limitations of the study (4.5), I suggest a more concise paragraph and that can be cited in conclusion as perspective. It will highlight your results and not present as limitations.

Reviewer 2 Report

Comments and Suggestions for Authors

The manuscript “Molecular Characterization and Opportunities for the Valorization of Traditional Olive Tree Varieties in Alto Aragon (NE Spain)” presents the studies which genotyped a total of 92 samples of 56 varieties/cultivars from 39 locations throughout the Alto Aragon, identified 29 new varieties and deposited in IFAPA's World Germplasm Bank of Olive Varieties. Then 8 single-varietal olive oils from Alto Aragon varieties were produced and characterized, and their organoleptic properties were evaluated. Furthermore, ancient olive trees were selected and 3D scanned to promote their protection as singular or monumental trees and for oleo-tourism purposes. The reported findings highlight the rich olive growing heritage of this northernmost frontier of olive tree cultivation in Spain. The studies are beneficial to help preserve genetic diversity and contribute to sustainable agriculture practices. Some minor revisions and following suggestions are made to improve the article.

1. There was no the section 4.4, straightly to the section 4.5 from 4.3.

2. In the section 3.3, maturity index (MI) significantly influenced olive oil extraction yield, physicochemical parameters and sensory attributes, which has been reported in many studies. It is recommended that eight single-variety olive oil were compared with the characteristics based on their similar MI in the present and previous works.

3. In the section 4.1, 29 previously undiscovered traditional olive varieties offered valuable genetic resources for olive breeding program. It is recommended to add a sentence at the end that these local varieties are a valuable source of diversity available for breeding, given that they generally grow under difficult agroclimatic conditions.

Reviewer 3 Report

Comments and Suggestions for Authors

Comments on the Quality of English Language

Moderate editing of the English language is required. It is recommended that the manuscript be submitted to a native speaker.

Reviewer 4 Report

Comments and Suggestions for Authors

Dear authors,

the submitted paper entitled:  Molecular Characterization and Opportunities for the Valorization of Traditional Olive Tree Varieties in Alto Aragon, describes the recovery and characterization of genotypes located in a particular territory of the Spain. The topic is very interesting, due to the current situation that requires more efforts to preserve the olive biodiversity.

However there are some aspects of the paper that need a revision.

The paragraph 3.2 is not clear. Please check it and try to explain the steps that you followed to define the list of genotypes. Are there reference cultivars? Only the two reported: Frantoio and Picual? It is not claear the experimental design for the PCoA and population structure. The legends of the figures must revised and better explained. I don't understand which are the 90 samples that you used for these two analyses, if you write that you have (58 or 46?) non redundant genotypes, why these 90? The PCoA just indicated that is totally missing a cluterization and the comment on this analysis is missing. Why did you perform a structure analysis? in my opinion the Structure is not applicable with your data, otherwise could be performed if you try to compare traditional Spanish cultivars with those sampled in this region. If you can access to molecular data of spanish cultivars you can perform this comparison. You can follow the process reported in the paper by Miazzi et al., 2020 Re.Ger.O.P.: An Integrated Project for the Recovery of Ancient and Rare Olive Germplasm (Frontiers in Plant Science 11:73.)

About the olive oil analyses, why you choose only these four genotypes among the new ones that you analyzed? Try to explain

Round 2

Reviewer 4 Report

Comments and Suggestions for Authors

Dear authors,

you performed a deep revision of the manuscript and the cover letter gives a point by point answering.

In my opinion the manuscript is more clear and better explained.

It could be suitable for the publication.